# Relation of Procollagen Type III Amino Terminal Propeptide Level to Sepsis Severity in Pediatrics

**DOI:** 10.3390/children8090791

**Published:** 2021-09-10

**Authors:** Nagwan Y. Saleh, Hesham M. Aboelghar, Sherif S. Salem, Shimaa E. Soliman, Doaa M. Elian

**Affiliations:** 1Pediatric Department, Faculty of Medicine, Menoufia University Hospital, Shebin El Kom 32511, Egypt; haboelghar@yahoo.com (H.M.A.); sss411@yahoo.com (S.S.S.); doaaelian@yahoo.com (D.M.E.); 2Medical Biochemistry and Molecular Biology Department, Faculty of Medicine Menoufia University, Shebin El Kom 32511, Egypt; dr.shelshafey2010@yahoo.com; 3Medical Biochemistry Unit, Department of Pathology, College of Medicine, Qassim University, Qassim 51452, Saudi Arabia; 4Pediatric Department, College of Medicine, King Faisal University, Al-Ahsa 31982, Saudi Arabia

**Keywords:** mortality, pediatric, PIIINP, shock, sepsis

## Abstract

Background: Sepsis is still the main etiology of mortality in pediatric intensive care units (PICUs). Therefore, we performed this study to evaluate the value of procollagen Type III amino-terminal propeptide (PIIINP) as a biomarker for sepsis severity diagnosis and mortality. Method: A prospective study was carried out on 170 critically ill children admitted into the PICU and 100 controls. The performed clinical examinations included calculation of the pediatric risk of mortality. Serum PIIINP was withdrawn from patients at admission and from the controls. Results: PIIINP level was significantly more increased in sepsis, severe sepsis, and septic shock than among the controls (*p* < 0.001). PIIINP was significantly higher in severe sepsis and septic shock (568.3 (32.5–1304.7) and 926.2 (460.6–1370), respectively) versus sepsis (149.5 (29.6–272.9)) (*p* < 0.001). PIIINP was significantly increased in non-survivors (935.4 (104.6–1370)) compared to survivors (586.5 (29.6–1169)) (*p* < 0.016). ROC curve analysis exhibited an area under the curve (AUC) of 0.833 for PIIINP, which is predictive for sepsis, while the cut-off point of 103.3 ng/mL had a sensitivity of 88% and specificity of 82%. The prognosis of the AUC curve for PIIINP to predict mortality was 0.651; the cut-off of 490.4 ng/mL had a sensitivity of 87.5% and specificity of 51.6%. Conclusions: PIIINP levels are increased in sepsis, with significantly higher levels in severe sepsis, septic shock, and non-survivors, thus representing a promising biomarker for pediatric sepsis severity and mortality.

## 1. Introduction

Sepsis is a syndrome characterized by a systemic inflammatory response to infection. Its incidence is very high, with a mortality rate from severe sepsis around 30–60% [1]. Epidemiological research has shown, relative to clinical data, that the incidence of sepsis in children accounts for approximately 8% of all admissions to pediatric intensive care units (PICU) [2].

Sepsis advocacy organizations play a vital role in improving innovative approaches to advance sepsis care for adults and children [3]. New York and Illinois have developed guidelines for all hospitals to follow sepsis protocols and achieve compliance in response to catastrophic cases of unrecognized sepsis in children [4,5]. Recently, 50 children’s hospitals in the United States joined a collaboration to improve sepsis outcomes, with the purpose of reducing sepsis deaths in children by 75% [6].

The reaction of host defenses to the stimulus is composed of simultaneous activation of a reactive network of inflammation, coagulation, and tissue repair that interact to increase the chance of host survival [7]. The repair of tissue requires angiogenesis, growth of epithelium, fibroblast migration and proliferation, and fibrogenesis. A standardized response to connective tissue repair was demonstrated in several organs after various types of damage [8]. Through tissue repair, healthy and segmented matrix components are released into extracellular fluid and circulation [8].

Collagens constitute about 30% of the protein body content and are responsible for strengthening the structure and elasticity of human tissue [9]. Type III collagen is present in the skin, the walls of blood vessels, and hollow viscera. Type III procollagen (PIIINP) is most commonly used as a marker of Type III collagen synthesis in early wound repair, but increased concentrations of PIIINP were also described in various disease states [10].

Procollagen type III amino-terminal propeptide (PIIINP), cleaved from the procollagen precursor molecule by specific proteins in the extracellular space, can be used as a biomarker for collagen synthesis [11]. Many studies have demonstrated that increased PIIINP levels in plasma reflect collagen synthesis at the disease site and can be used as an indicator of a reparative process, independent of etiology [8,12,13].

N-terminal propeptide of type I procollagen (PINP) is recommended as a marker of bone resorption and bone formation and is related to corresponding histomorphometric bone formation measures and resorption [14]. Therefore, this study focused on PIIINP and not PINP.

There is increasing evidence that PIIINP is elevated early in septic episodes and can indicate the acute organ dysfunction or failure characterizing severe sepsis [15]. Nevertheless, no study has investigated the diagnostic performance of PIIINP in pediatric sepsis. Therefore, we performed the present prospective cohort study to assess the role of PIIINP as a marker for sepsis severity diagnosis and mortality in critically ill children.

## 2. Subjects and Methods

### 2.1. Design

This is a prospective study carried out on 170 children admitted into a 10-bed pediatric intensive care unit (PICU) at Menoufia University Hospital, Egypt, and 100 apparently healthy controls of matched age and sex from June 2018 to May 2019. We included children who fulfilled the following criteria: (1) children aged from 1 month up to 18 years, and (2) with a critical illness requiring ICU admission. Exclusion criteria were (1) patients in the neonatal period or those older than 18 years, and (2) patients who were not available for follow-up for 30 days after discharge. Another group, comprised of 100 children of matched age and sex, was chosen as the control group. They were healthy children who were not hospitalized and with no pathological findings recorded on examinations. Informed consent was taken from the parents of the included subjects (patients and controls) in the study.

The studied patients were divided into three groups according to the international pediatric sepsis consensus conference as sepsis, severe sepsis, and septic shock [16]. Sepsis is the systemic response to infection manifested by two or more of the following, resulting in infection: (a) a temperature of more than 38 or less than 36 °C, (b) heart rate > 90 beats/min, (c) respiratory rate > 20 breaths per minute or PaCO_2_ < 32 mm Hg, and (d) white blood cell count (WBC) of more than 12,000/mm^3^ or less than 4000/mm^3^, or more than 10% immature (band) forms. Severe sepsis is sepsis accompanied by organ dysfunction, poor perfusion, or sepsis-induced hypotension. Septic shock is a subgroup of severe sepsis that is diagnosed as sepsis-induced hypotension, persisting despite adequate fluid resuscitation.

Organ failure is defined as respiratory, cardiovascular, central nervous system failure, and hepatic, renal, hematologic, and metabolic failure [17]. Pediatric multiple-organ dysfunction syndrome (P-MODS) is the dysfunction of more than two organs. The disseminated intravascular coagulation (DIC) score is a score of 5 or more according to the International Society on Thrombosis and Hemostasis subcommittee guidelines [16].

The primary outcome measure was the occurrence of death during hospital admission or through a follow-up period of 30 days after hospital discharge. Secondary outcome measures included the length of PICU stay and stay in the hospital, the need for and duration of mechanical ventilation, vasopressor use, pediatric multiple-organ dysfunction (P-MOD), and DIC scores.

### 2.2. Methods

For each patient, a complete diagnostic workup was performed, including a thorough history and physical examination. Physical examination included recording the heart rate, respiratory rate, systolic and diastolic blood pressures, pupillary reaction, and Glasgow Coma Scale. The workup also comprised arterial blood gases, random blood glucose, and complete blood count (CBC) using a Pentra-80 automated blood counter (ABX–Franc–Rue du Caducee-Paris Euromedecine-BP-7290.34184 Montpellier-Cedex4). C-reactive protein (CRP), serum electrolytes, liver function tests (ALT and AST by kinetic UV optimized method IFCC (LTEC Kit, Holliston, MA, USA), kidney function tests (blood urea and serum creatinine by colorimetric methods using DIAMOND diagnostics kits, Germany), prothrombin time (PT) by computerized coagulation analyzer and blood culture. In addition, chest radiograph, brain CT, and other laboratories or radiological investigations were performed when appropriate. Two severity scores were calculated, namely the Pediatric Risk of Mortality (PRISM) [18] and Pediatric Index of Mortality 2 (PIM2) [19]. PRISM score is automatically calculated within 24 h of admission. PIM2 is simpler and needs to be calculated within 1 h of face-to-face contact with the patient. This automatically yielded the predicted death rates.

A single PIIINP measurement was performed for all patients within 24 h of admission in the PICU and for the controls. Serum PIIINP was determined using an ELISA kit bought from Glory Science Co., Ltd., Del Rio, TX, USA, catalog no. 201-12-1354. The kit uses a double-antibody sandwich enzyme-linked immunosorbent assay (ELISA) to assay the level of human N-terminal procollagen III propeptide (PIIINP) in samples. N-terminal procollagen III propeptide (PIIINP) was added to a monoclonal antibody enzyme well, which was precoated with human N-terminal procollagen III propertied (PIIINP) monoclonal antibodies.

#### Ethical Approval

All procedures performed during the study were in accordance with the ethical standards of the Menoufia University Institutional Research Committee, which approved the study (ID: 11/2019.PED). All parents gave their informed consent for their children to participate in the study.

### 2.3. Statistical Analysis

Data are represented as mean ± SD, median, and range. Categorical data were analyzed by chi-squared test. Continuous variables were compared by *t* test. The Mann–Whitney U test was used for continuous variables with skewed distribution and post hoc analysis when the groups were small. The Kruskal–Wallis test was used for comparisons among more than two groups. Correlations between variables were analyzed using Pearson’s correlation. The diagnostic powers of the PIIINP and other variables tested by the receiver operating characteristic (ROC) curve with the Youden index were used to select the optimal cut-off values. Analysis was performed using IBM SPSS software version 20.0 (SPSS, Chicago, IL, USA). Two-sided *p* values of less than 0.05 were considered to be significant.

## 3. Results

### 3.1. Demographic and Clinical Characteristics of the Studied Groups

A total of 170 critically ill children with 100 apparently healthy controls enrolled in this study. The median age of the included patients was 26.8 months (range, 1.5–192 months) and consisted of 75 males and 95 females; the median age was 28.8 months (range, 2–180 months; 40 males and 60 females) for the matched controls. Of the patients, 19 children (11.2%) died (Table 1).

### 3.2. Studied Patients Regarding Severity and Outcome of the Disease

Of the children, 90 (52.9%) had sepsis, 53 (31.2%) had severe sepsis, and 27 (15.9%) had septic shock. The primary causes of sepsis in patients were respiratory (45%), gastrointestinal (18.9%), cardiac (13.8%), neurological (9.7%), metabolic (4.7%), renal (3.4%), surgical (2.3%), and hematological (2.2%) disorders. Pathogenic bacteria were isolated from the blood culture in 127 of 170 septic children (75%) with 65% Gram-negative, 15% Gram-positive bacteria, and 20% mixed infections. They received antibiotics in the form of vancomycin, piperacillin/tazobactam, ceftriaxone, cefepime, and meropenem. Of the children, 27 (15.9%) received mechanical ventilation, and 15 (15.9%) needed inotropic drugs. There were highly significant differences in the severity of disease among sepsis groups on admission as determined by PRISM, P-MODS, and DIC scores (*p* < 0.001) (Table 2).

### 3.3. Outcome of the Studied Groups

The duration of PICU and hospital stay was 5.6 (4 ± 10) and 16 (9 ± 21) days, respectively, in the septic group, 9.5 (8 ± 17) and 17 (14 ± 33) days, respectively, in the severe septic group, and 9 (4 ± 16) and 18 (12 ± 28) days, respectively, in the septic-shock group; there were no significant differences among them. Mortality occurred in 19 (11.2%) children; 16 (84.2%) of them had had septic shock. Mortality was significantly increased in the septic shock group compared to in the septic and severe septic groups (*p* = 0.018; Table 2).

### 3.4. Association of Serum PIIINP with Disease Severity and Mortality

Serum PIIINP level was significantly more elevated among total patients and patients with sepsis, severe sepsis, and septic shock compared to that among controls (*p* < 0.001). Serum PIIINP level showed a significant increase in patients with severe sepsis and septic shock (568.3 (32.5–1304.7)) and (926.2 (460.6–1370)) ng/mL, respectively, versus sepsis (149.5 (29.6–272.9)) (*p* < 0.001). PIIINP level was increased in mechanically ventilated patients (908.3 (104.6–1370)) compared with that of the no mechanically ventilated subgroup (578.9 (29.6–1304.1)) (*p* < 0.029). In addition, PIIINP was significantly increased in non-survivors (935.4 (104.6–1370)) compared to in survivors (586.5 (29.6–1169)) (*p* < 0.016; Table 3).

### 3.5. Correlations of PIIINP with Other Clinical and Laboratory Parameters

PIIINP had a significant positive correlation with hospital stay, PRISM, PIM2 mortality risk percentages, CRP, WBCs and PT. In addition, PIIINP was significantly correlated with PICU stay, P-MODS, and DIC scores. On the other hand, PIIINP was not correlated with age, creatinine, albumin, platelet, bilirubin and fibrinogen (Table 4).

### 3.6. Area under the ROC Curve for PIIINP in Predicting Sepsis and Mortality

ROC curve analysis exhibited an area under the curve (AUC) of 0.833 for PIIINP that predicted patients with sepsis; the cut-off point of 103.3 ng/mL had a sensitivity of 88% and specificity of 82% (Table 5 and Figure 1).

Regarding prognosis, the AUC curve for PIIINP to predict mortality was 0.651; the cut-off of 490.4 ng/mL had a sensitivity of 87.5% and specificity of 51.6% (Table 6 and Figure 2).

## 4. Discussion

The present study tested PIIINP as a valuable biomarker of sepsis in the pediatric ICU. We noticed a highly significant elevation in serum PIIINP in the patient groups of sepsis, severe sepsis, and septic shock compared to in the control group, and serum PIIINP was significantly increased in the severe sepsis and septic shock groups compared to in the sepsis group. These results provide further support for the usefulness of PIIINP in assessing the severity of sepsis in children. Furthermore, PIIINP proved through ROC curve analysis to have excellent power for sepsis prediction; a cut-off level of PIIINP of 103.3 ng/mL had sensitivity and specificity of 88% and 82%, respectively. To the best of our knowledge, this is the first study that assessed the diagnostic performance of PIIINP in the setting of pediatric sepsis; however, previous reports have assessed serum PIIINP level in neonatal and adult sepsis.

In agreement with our findings, Said et al. [20] assessed levels of PIIINP among preterm neonates and found that serum PIIINP was significantly higher in septic cases than in the controls. The highest levels were in the ventilated group, followed by the septic and grower groups. Similarly, Zakynthinos et al. [15] tested the hypothesis that PIIINP is elevated early in septic episodes and indicated the acute organ dysfunction or failure characterizing severe sepsis in adults. PIIINP increased in patients with sepsis, exhibiting further significant increases in patients with severe sepsis and septic shock. Gäddnäs et al. [10] showed that PIIINP concentration was increased in septic patients when compared with controls on day 1.

Elevated serum PIIINP levels in sepsis patients are associated with increased hepatic synthesis or reduced hepatocyte uptake because the liver eliminates procollagen peptides. In addition, although the kidneys secrete small amounts of PIIINP, acute renal failure is accompanied with elevated synthesis of type III collagen [15].

Previous studies have reported on the metabolism of collagen in severe trauma, acute respiratory distress syndrome (ARDS), or Gram-negative sepsis [12,21,22]. The level of PIIINP increases during the process of disease, and PINP increases in the late stage with the development of multiple-organ failure (MOF) and mortality.

Collagen synthesis in the lungs has been studied in critical illness. Acute respiratory distress syndrome is the most serious manifestation of acute lung injury and one of the most common organ failures in severe sepsis. Collagen I and III peptides are increased in plasma and bronchoalveolar lavage fluid in patients with acute respiratory distress syndrome during the first days of illness and are accompanied by an increased risk of mortality [21,23,24].

Regarding mechanical ventilation, there was a significant elevation in PIIINP level among mechanically ventilated patients compared to patients who were not mechanically ventilated (*p* = 0.029). In addition, we found a significant difference between survivors and non-survivors (*p* = 0.016). Moreover, we assessed the prognostic value of PIIINP, and we noticed a significant elevation in the level of PIIINP at admission among critically ill children who subsequently died when compared with those who survived, in line with the findings of this study [15].

In our results, there was significant correlation among length of hospital stay, PICU stay, P-MODS and DIC scores, and PIIINP level. There was also significant correlation between PIIINP level and elevated mortality scores. In line with our findings, Gäddnäs et al. [10] reported that maximal serum PIIINP concentrations during sepsis were higher in non-survivors compared with survivors, and in multiple organ failure (MOF) compared with multiple organ dysfunction syndrome (MODS). In addition, Said et al. [20] and Zakynthinos et al. [15] reported similar findings. In our research, there were positive correlations between PIIINP and CRP (r = 0.327; *p* = 0.004) and WBCs (r = 0.260; *p* = 0.024); their ability was studied with the ROC curve for the prediction of sepsis and showed an AUC for PIIINP equal to 0.883 with a combined sensitivity and specificity of 88% and 82% with a cut-off point of 103.3 ng/mL. The AUC for CRP was equal to 0.986 with a combined sensitivity and specificity of 88% and 99.2% with a cut-off point of 10.22, and the AUC for WBCs was equal to 0.880 with a combined sensitivity and specificity of 81% and 56% with a cut-off point of 6.9. Zakynthinos et al. [15] reported that the AUC for PIIINP was 0.96, and C-reactive protein and leukocyte count also exhibited high diagnostic accuracy to distinguish control subjects from patients with septic conditions with AUCs of 0.99 and 0.91, respectively.

Regarding assessing mortality scores in our research, there were positive correlations between PIIINP and PRISM scores (r = 0.313; *p* = 0.047) and PIM II (r = 0.315; *p* = 0.044), and its ability as studied by a ROC curve showed an AUC equal to 0.65 with a combined sensitivity and specificity of 88% and 82% with a cut-off point of 490.4 for PIIINP for mortality prediction. The AUC for PRISM mortality risk was equal to 0.93 with a combined sensitivity and specificity of 87.5% and 78% with a cut-off point of 4.25 for prediction of mortality. The AUC for PIM II mortality risk was equal to 0.96 with a combined sensitivity and specificity of 99% and 89% with a cut-off point of 6.8 for mortality prediction.

Procollagen type III amino-terminal propeptide (PIIINP), cleaved from the procollagen molecule precursor by specific proteinases in the extracellular space, is used as a biological marker of collagen synthesis, and various studies have shown that increased plasma PIIINP levels reflect collagen synthesis at the site of disease and can be used as biomarkers of a reparative process independent of etiology. Sepsis (i.e., infection and the systemic inflammatory response to it) represents extensive tissue injury that usually begins with connective tissue repair [15].

There is elevation in serum levels of procollagen Type III propeptide in severely injured patients, and this is accompanied with multiple-organ failure (MOF) and death. Therefore, the more severe the sepsis is, the more PIIINP is elevated, and this gives PIIINP prognostic value in sepsis [10]. Waydhas et al. [10] reported increased PIIINP serum concentrations in severely injured patients. Similar to our findings in septic patients, serum concentration was increased in severely injured non-survivors and in those who had developed MOF. This agrees with the results reported by Said et al. [20] and Zakynthinos et al. [15].

Increased levels of procollagen III amino-terminal propeptide (PIIINP) in bronchoalveolar lavage fluid (BAL) can be detected in the early stages of acute respiratory distress syndrome. Serum PIIINP levels correlate with disease severity and death risk in severe sepsis; serum PIIINP concentrations are increased, and maximal levels are higher in multiple organ failure and in non-survivors [25].

A limitation of this study is that the authors did not sequentially measure PIIINP to track changes in level in response to treatment

## 5. Conclusions

In critically ill children, PIIINP level on admission is a valuable biomarker for diagnosing sepsis severity and mortality. Larger studies are necessary to fully assess the role of PIIINP in children.

## Figures and Tables

**Figure 1 children-08-00791-f001:**
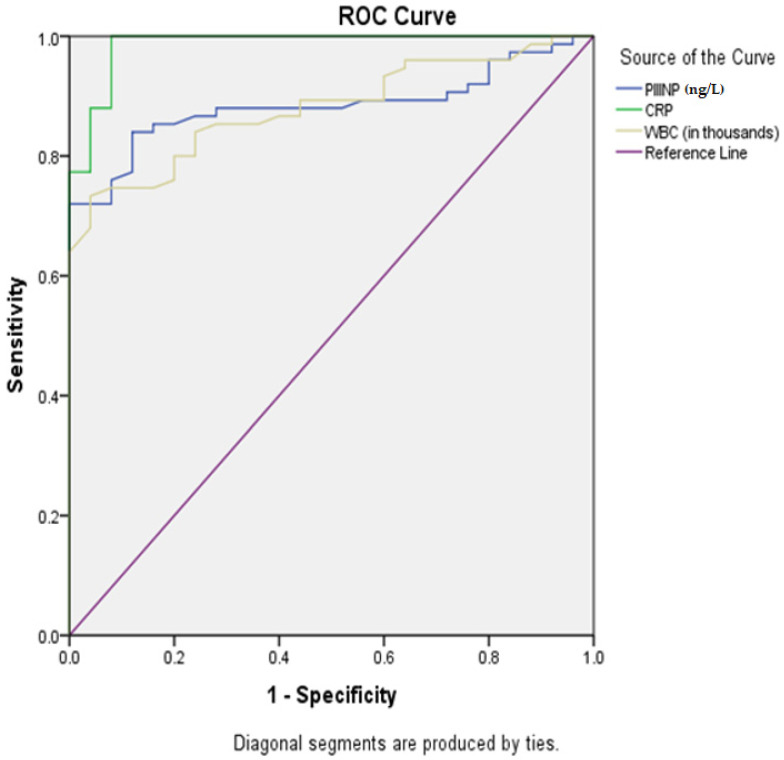
Receiver-operating characteristic (ROC) curve for prediction of sepsis severity.

**Figure 2 children-08-00791-f002:**
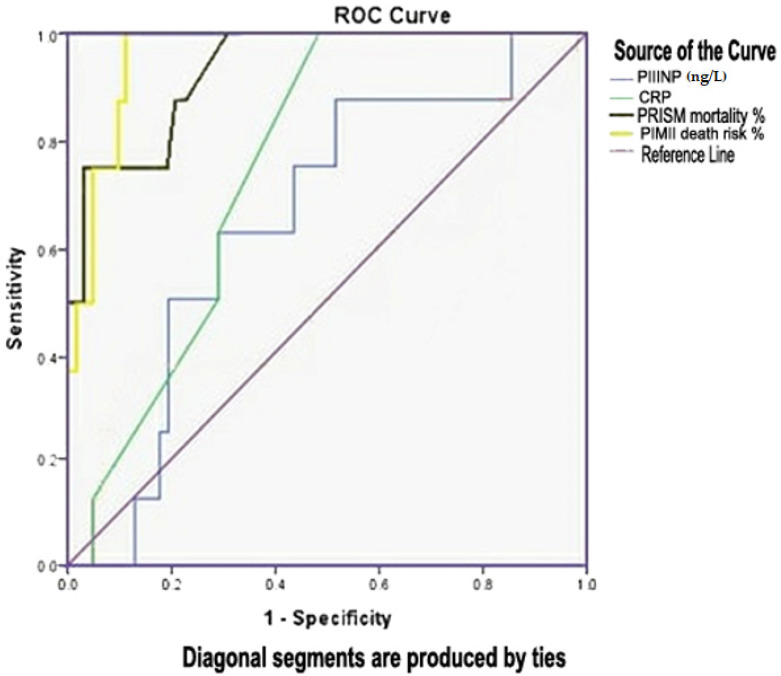
Receiver-operating characteristic (ROC) curve for prediction of mortality.

**Table 1 children-08-00791-t001:** Demographic and clinical characteristics of the studied groups.

Studied Variables	The Studied Groups		Test of Significance	*p* Value
	Patients (*n* = 170)	Control (*n* = 100)		
Age (month)				
Mean ± SD	26.53 ± 40.2	28.9 ± 44.7		
Median (Range)	26.8 (1.5–192)	28.8 (2–180)	U = 0.2578	0.79
Sex No (%)				
Male	75 (44.2%)	40 (40.0%)		
Female	95 (55.8%)	60 (60.0%)	*Χ^2^* = 0.81	0.81
Weight (Kg)				
Mean ± SD	10.05 ± 7.7	11.7 ± 3.12		
Median (Range)	10.5 (2.1–45)	11.5 (4.2–40)	U = 0.7618	0.44
Height (cm)				
Mean ± SD	78.19 ± 24.3	75.1 ± 20.9		
Median (Range)	88.12 (45–161)	77.2 (55–130)	U = 0.5837	0.56
BMI (kg/m^2^)				
Mean ± SD	14.7 ± 2.8	14.8 ± 2.1		
Median (Range)	14.9 (8.2–21.6)	15.6 (8.7–22.1)	U = 0.1670	0.86
MV No (%)				
Yes	27 (15.9%)	-	-	-
No	143 (84.1%)			
MV (h)				
Mean ± SD	276.6 ± 298.2	-	-	-
Range	55–1028			
PICU stay/day				
Mean ± SD	7.85 ± 9.2			
Range	1–16	-	-	-
Hospital stay/day				
Mean ± SD	10.36 ± 9.7			
Median (Range)	10.50 (2–35)	-	-	-
PRISM score				
Mean ± SD	22.1 ± 15.2			
Median (Range)	27.5 (0–46)	-	-	-
PRISM mortality risk %				
Mean ± SD	25.63 ± 28.9			
Median (Range)	25.72 (8–64.4)	-	-	-
PIM mortality risk %				
Mean ± SD	6.85 ± 12.4			
Median (Range)	6.96 (2–66)	-	-	-
Mortality No (%)				
Yes	19 (11.2%)			
No	151 (88.8%)	-	-	-

PRISM = Pediatric Risk of Mortality; PIM = Pediatric Index of Mortality; MV = Mechanical ventilation; U = Mann Whitney U test; X^2^ = Chi-squared.

**Table 2 children-08-00791-t002:** Demographic, disease severity and outcome data in studied groups.

	Sepsis (*n* = 90)	Severe Sepsis (*n* = 53)	Septic Shock (*n* = 27)	*p* Value
Age (Months)				
Median (Range)	58.9 (25.2 ± 121.2)	20.4 (2.4 ± 43.2)	81.6 (55.2 ± 175.2)	0.076
Sex No				
Male	35	28	12	0.934
Female	55	25	15	
Disease severity				
MV (*n* = 27)	3	5	19	<0.001 **
Use of vasopressor	0	0	27	<0.001 **
PRISM Median (Range)	9 (6 ± 10)	15 (9 ± 17)	25 (19 ± 46)	<0.001 **
P-MODS	0 (0 ± 1)	1 (0 ± 3)	6 (3 ± 9)	<0.001 **
DIC score	2 (0 ± 2)	3 (1 ± 3)	5 (3 ± 5)	<0.001 **
Outcome				
Mortality (*n* = 19)	1	2	16	0.018 *
PICU stay (days)				
Median (Range)	5.6 (4 ± 10)	9.5 (8 ± 17)	9 (4 ± 16)	0.137
Hospital stays				
Median (Range)	16 (9 ± 21)	17 (14 ± 33)	18 (12 ± 28)	0.562

PRISM = Pediatric Risk of Mortality; P-MODS = Pediatric-Multiple Organ Dysfunction Score; DIC = disseminated intravascular coagulation; PICU = pediatric intensive care unit; MV = mechanical ventilation; * = significant; ** = highly significant.

**Table 3 children-08-00791-t003:** Serum PIIINP level in the studied groups, subgroups, and controls.

PIIINP (ng/mL)	The Studied Groups		Mann Whitney (U) Test	*p* Value
	Patients (*n* = 170)	Control (*n* = 100)		
PIIINP				
Mean ± SD	614.9 ± 432.9	3.6 ± 0.66	6.7826	<0.001 **
Median (Range)	615.9 (29.5–1370)	3.9 (1.9–4.9)		
	Sepsis (*n* = 90)	Control (*n* = 100)		
PIIINP				
Mean ± SD	149.24 ± 79.79	3.6 ± 0.66	6.8953	<0.001 **
Median (Range)	149.5 (29.6–272.9)	3.9 (1.9–4.9)		
	Severe sepsis (*n* = 53)	Control (*n* = 100)		
PIIINP				
Mean ± SD	568.26 ± 440.8	3.6 ± 0.66	5.391	<0.001 **
Median (Range)	568.3 (32.5–1304.7)	3.9 (1.9–4.9)		
	Septic shock (*n* = 27)	Control (*n* = 100)		
PIIINP				
Mean ± SD	926.16 ±246.7	3.6 ± 0.66	7.1568	<0.001 **
Median (Range)	926.2 (460.6–1370)	3.9 (1.9–4.9)		
	Sepsis (*n* = 90)	Severe sepsis (*n* = 53)		
PIIINP				
Mean ± SD	149.24 ± 79.79	568.26 ± 440.8	5.5085	<0.001 **
Median (Range)	149.5 (29.6–272.9)	568.3 (32.5–1304.7)		
	Sepsis (*n* = 90)	Septic shock (*n* = 27)		
PIIINP				
Mean ± SD	149.24 ± 79.79	926.16 ± 246.7	2.1472	<0.001 **
Median (Range)	149.3 (29.6–272.9)	926.2 (460.6–1370)		
	Severe sepsis (*n* = 53)	Septic shock (*n* = 27)		
PIIINP				
Mean ± SD	568.26 ± 440.8	926.16 ± 246.7	3.9023	<0.001 **
Median (Range)	568.3 (32.5–1304.7)	926.2 (460.6–1370)		
	Mechanically ventilated (*n* = 27)	Non-mechanically ventilated (*n* = 143)		
PIIINP				
Mean ± SD	908.2 ± 206.85	578.86 ± 431.57	2.228	0.029 *
Median (Range)	908.3(104.6–1370)	578.9 (29.6–1304.1)		
	Survivors (*n* = 151)	Non-Survivor (*n* = 19)		
PIIINP				
Mean ± SD	586.46 ± 433.7	935.32 ± 232.9	2.229	0.016 *
Median (Range)	586.5 (29.6–1169)	935.4 (104.6–1370)		

U = Mann Whitney U test; * = significant; ** = highly significant.

**Table 4 children-08-00791-t004:** Correlation between PIIINP level and other clinical parameters in the studied groups.

Studied Variable	Serum PIIINP	
	R	*p* Value
Age	0.207	0.074
Hospital stay	0.251	0.03 *
PICU stay	0.318	<0.001 **
PRISM risk mortality %	0.313	0.047 *
PIM risk mortality %	0.315	0.044 *
WBCs	0.26	0.024 *
Platelets	0.064	0.58
CRP	0.327	0.004 *
Total bilirubin	−0.045	0.703
Albumin	−0.105	0.37
Creatinine	0.01	0.93
PT	0.479	0.003 *
Fibrinogen	−0.288	0.069
DIC score	0.562	<0.001 **
P-MODS	0.551	<0.001 **

PICU stay = pediatric intensive care unit stay; PRISM = Pediatric Risk of Mortality; PIM = Pediatric Index of Mortality; WBCs = white blood cells; CRP = C-reactive protein; PT = prothrombin time; P-MODS = Pediatric Multiple Organ Dysfunction Score; DIC = disseminated intravascular coagulation; * = significant; ** = highly significant; R = Pearson correlation.

**Table 5 children-08-00791-t005:** Validity of PIIINP, CRP, and WBCs for prediction of sepsis.

	PIIINP	CRP	WBCs
AUC	0.883	0.986	0.88
*p* value	<0.001	<0.001	<0.001
95% CI	0.81–0.94	0.96.5−0.99	0.815–0.945
Cutoff point	103.3	10.22	6.9
Sensitivity	88%	88%	81%
Specificity	82%	99.20%	56%
PPV	83%	90%	66%
NPV	87%	89%	82%
Accuracy	88%	90%	67.50%

PIIINP = procollagen type III amino terminal propeptide; CRP = C-reactive protein; WBCs = white blood cells; AUC = area under curve; PPV = positive predictive value; NPV = negative predictive value.

**Table 6 children-08-00791-t006:** Validity of PIIINP, CRP, PRISM and PIM for prediction of mortality.

	PIIINP	CRP	PRISM Mortality Risk	PIM II Mortality Risk
AUC	0.651	0.749	0.93	0.96
*p* value	<0.001	<0.001	<0.001	<0.001
95% CI	0.47–0.83	0.62–0.87	085–0.99	0.915–0.99
Cutoff point	490.4	41.4	4.25	6.8
Sensitivity	87.50%	99%	87.50%	99%
Specificity	51.60%	52%	78%	89%
PPV	16.70%	18.60%	76.70%	89%
NPV	97%	99.70%	87%	99%
Accuracy	88%	90%	88%	99%

PIIINP = procollagen type III amino terminal propeptide; CRP = C-reactive protein; PRISM = Pediatric Risk of Mortality; PIM = Pediatric Index of Mortality; PPV = positive predictive value; NPV = negative predictive value.

## Data Availability

Data is available on request due to restrictions, e.g., privacy or ethical.

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
