# Peer review of "Relation of Procollagen Type III Amino Terminal Propeptide Level to Sepsis Severity in Pediatrics"

_children, 2021, doi:10.3390/children8090791_

Round 1

Reviewer 1 Report

This article highlights procollagen type III amino-terminal propeptide (PIIINP) could be a promising marker for the diagnosis of sepsis severity and mortality in children. The authors provide details of the analysis and methodology. However, there are several typographical errors in should be addressed and be consistent.

Major concerns:

  1. In my opinion, the introduction seems too short and not persuasive. Introduction requires an extensive amount of work. Authors could provide current CDC reports on pediatric sepsis surveillance and provide an extensive background of collagen synthesis and degradation in sepsis.
  2. Grammar and spelling should be checked. For example, in line 28, “non survivor” should be “non-survivor” or “nonsurvivors”…same as ”non mechanically” in line 145. Line 120, sStudied?
  3. Citation (2) is not appropriate for the statement.
  4. The authors stated that “Many studies have demonstrated that increased PIIINP levels in plasma reflect collagen synthesis at the disease site…….”, however, only one reference was cited here. Authors may provide convincing evidence regarding your statement since this is a highlight of your work.
  5. Although table 2 showed data clearly, result 3.3 is difficult to read. Authors should write this result in an understandable way. This is also one of the reasons that I would recommend this paper can be revised by a native speaker.
  6. How did the authors exclude the possibility that collagen propeptide I level would be associated with pediatric sepsis severity and mortality?
  7. What are the “PPV” and “NPP” in table 6? Authors should give their full name under the table and in the context.
  8. For the discussion section, there are no attempts to discuss the relationship between collagen turnover and inflammatory responses in sepsis.

Minor concerns:

  1. Any reagent the authors listed should have a catalog number.
  2. Line 78- the abbreviation DIC should have a full name if the authors want to mention it here for the first time. What is the “MV” in table 2? You may give it a full name below the table or mention it when you first state this term. Similarly, WBC, PT and CRP in table 4 should have their full name in the method since authors have mentioned their full names but I cannot see abbreviations in the method.
    Line 116, receiver operating characteristic should have an abbreviation, ROC.
    The same issue in Line 182 - ARDS should have a full name “Acute Respiratory distress syndrome” before line 186.

At the current state, this manuscript needs major revisions for the grammar and syntax and for contents.

Author Response

Reviewer response

Reviewer 1

This article highlights procollagen type III amino-terminal propeptide (PIIINP) could be a promising marker for the diagnosis of sepsis severity and mortality in children. The authors provide details of the analysis and methodology. However, there are several typographical errors in should be addressed and be consistent.

Major concerns:

Comment 1. In my opinion, the introduction seems too short and not persuasive. Introduction requires an extensive amount of work. Authors could provide current CDC reports on pediatric sepsis surveillance and provide an extensive background of collagen synthesis and degradation in sepsis.

Answer (1): Yes, Sir. I added as you requested in the introduction section and marked that by red line. As follow:

Sepsis advocacy organizations have played an important role to develop innovative approaches to improve sepsis care for adults and children [3].New York and Illinois, established regulations for all hospitals to implement sepsis protocols and track compliance in response to tragic cases of unrecognized sepsis in children [4,5]. More recently, 50 US children’s hospitals joined the Improving Pediatric Sepsis Outcomes collaborative, this aims to reduce pediatric sepsis mortality by 75% [6].

Collagens constitute about 30% of the body’s protein content and responsible for strength of the structure and elasticity of human tissue [9]. Type III collagen is present in skin, walls of blood vessel and hollow viscera. Type III procollagen (PIIINP) is most commonly used as a marker of type III collagen synthesis in early wound repair, but increased concentrations of PIIINP have also been described in various disease states [10].

Comment 2. Grammar and spelling should be checked. For example, in line 28, “non survivor” should be “non-survivor” or “nonsurvivors”…same as ”non mechanically” in line 145. Line 120, sStudied?

Answer (2): Yes, Sir. I corrected them through all the manuscript by red line.

Comment 3-Citation (2) is not appropriate for the statement.

Answer (3): Yes, Sir. I changed this statement and added another one with accurate citation in the manuscript by red line as follow:

Epidemiologic studies using clinical data have found an incidence of pediatric sepsis in up to 8% of all pediatric intensive care unit (PICU) admissions [2].

Comment (4): The authors stated that “Many studies have demonstrated that increased PIIINP levels in plasma reflect collagen synthesis at the disease site…….”, however, only one reference was cited here. Authors may provide convincing evidence regarding your statement since this is a highlight of your work.

Answer (4): Yes, Sir. I added the other references in the manuscript by red line as follow:

13-Horslev-Petersen K. Circulating extracellular matrix components as markers for connective tissue response to inflammation. Dan Med Bull 1990; 37:308-29.

14-Kirk JE, Bateman ED, Haslam P, Laurent GJ, Turner-Warwick M. Serum type III procollagen peptide concentration in cryptogenicfibrosing alveolitis and its clinical relevance. Thorax 1984; 39: 726-32.

Comment (5): Although table 2 showed data clearly, result 3.3 is difficult to read. Authors should write this result in an understandable way. This is also one of the reasons that I would recommend this paper can be revised by a native speaker.

Answer (5): Yes, Sir. I already revised the paper by a native speaker in MDPI Billing <[email protected]>

.Comment (6): How did the authors exclude the possibility that collagen propeptide I level would be associated with pediatric sepsis severity and mortality?

Answer (5): Yes, Sir. We used kit that has Human N-terminal procollagen III propeptide (PIIINP) monoclonal antibody that highly specific to Human N-terminal procollagen III propeptide (PIIINP) in samples. We added that in the method section as below;

The kit uses a double-antibody sandwich enzyme-linked immunosorbent assay (ELISA) to assay the level of Human N-terminal procollagen III propeptide (PIIINP) in samples. Add N-terminal procollagen III propeptide (PIIINP) to monoclonal antibody Enzyme well which is pre-coated with Human N-terminal procollagen III propeptide (PIIINP) monoclonal antibody.

Comment (7): What are the “PPV” and “NPV” in table 6? Authors should give their full name under the table and in the context.

Answer (7): Yes, Sir. PPV means “Positive predictive value” and NPV means Negative predictive value and I write then in the manuscript by red line.

Comment (8): For the discussion section, there are no attempts to discuss the relationship between collagen turnover and inflammatory responses in sepsis.

Answer (8): Yes, Sir. I already added the relationship between collagen turnover and inflammatory responses in sepsis in the discussion section as below:

    The increased serum PIIINP levels in septic patients may relate to increased liver synthesis or decreased uptake by liver cells because procollagen propeptides are eliminated by the liver. In addition, although small amounts of PIIINP are excreted by the kidneys, acute renal failure is associated with increased synthesis of type III collagen [15].

Minor concerns:

  1. Any reagent the authors listed should have a catalog number.

Answer (1): Yes, Sir. I already added a catalogue No. 201-12-1354 in the manuscript.

  1. Line 78- the abbreviation DIC should have a full name if the authors want to mention it here for the first time. What is the “MV” in table 2? You may give it a full name below the table or mention it when you first state this term. Similarly, WBC, PT and CRP in table 4 should have their full name in the method since authors have mentioned their full names but I cannot see abbreviations in the method.
    Line 116, receiver operating characteristic should have an abbreviation, ROC.
    The same issue in Line 182 - ARDS should have a full name “Acute Respiratory distress syndrome” before line 186.

Answer (2): Yes, Sir. I corrected them in the manuscript by red line.

Reviewer 2 Report

The authors investigated the association between relation of procollagen type III amino terminal propeptide 2 level to sepsis severity in pediatrics. Although many studies investigated circulated biomarker to predict sepsis, we believed there are some interaction that we undiscovered. However, there are some problems had to be detail clarified 

1. We suggested authors to re-design the table. Authors should try to make the table more easily understand.

  1. The detail data of sepsis group has to report, including infection type, pathogen, antibiotics, ……etc
  2. There are several scores and biomarker analyzed in this article. But authors did not report the ROC curve.
  3. In current concept, we may use Neutrophil-Lymphocyte Ratio (NLR), CRP, PCT, WBC……. etc to predict sepsis. Author should analyze these basic parameter and compare between them. In addition, author should report the likehood ratio.
  4. There was missing subgroup analysis. Did the accuracy of PIIINP predicting sepsis be similar in different age? in different infection types (UTI, pneumonia, FUO)? in different pathogen (GNB, GPC….)?
  5. Did the reference range of PIIINP not be age-adjusted????
  6. In the section of discussion, it is necessary to report also all the new studies of the literature on this topic which are now missing. 
  7. The mechanism of pathophysiological links in this field is reported in many previous studies. We suggested authors to report current pathogenesis in article.
  8. Wu suggested author to provide a figure to overview results in this articles to make reader easily understand.

Author Response

Reviewer response

Reviewer 2

The authors investigated the association between relation of procollagen type III amino terminal propeptide levels to sepsis severity in pediatrics. Although many studies investigated circulated biomarker to predict sepsis, we believed there are some interactions that we undiscovered. However, there are some problems had to be detail clarified 
Comment 1. We suggested authors to re-design the table. Authors should try to make the table more easily understand.

Answer (1): Yes, Sir. I already re-design the tables.

Comment 2 The detail data of sepsis group has to report, including infection type, pathogen, antibiotics,etc

Answer (2): Yes, Sir. I added the detail data of sepsis group accordingly to infection type, pathogen, and antibiotics as you requested in the manuscript by red line as below;  

The primary reasons of septic patients included; respiratory (45%), gastrointestinal (18.9%), cardiac (13.8%), neurological (9.7%), metabolic (4.7%), renal (3.4%), surgical (2.3%), and hematological (2.2%) disorders. Pathogenic bacteria were isolated from blood culture in in 127 of 170 septic patients (75%) with 65% gram-negative, 15% gram positive bacteria and 20% mixed infections. They received antibiotic inform of Vancomycin, piperacillin/tazobactam, ceftriaxone, cefepime, and meropenem.

3- There are several scores and biomarker analyzed in this article. But authors did not report the ROC curve.

Answer (3): Yes, Sir. Roc curve is one of analytic statistical tests that used to determine the optimum cut off value for the studied diagnostic marker and the associated specificity and sensitivity level.  This is reported in Statistical analysis section.

4- In current concept, we may use Neutrophil-Lymphocyte Ratio (NLR), CRP, PCT, WBC……. etc to predict sepsis. Author should analyze these basic parameter and compare between them. In addition, author should report the likehood ratio.

  Answer (4): Yes, Sir. We added analysis of basic parameter CRP and WBCs in the manuscript.  But Neutrophil-Lymphocyte Ratio (NLR), PCT not done, this is considered as limitation of the study.

In our research ,we found that there was positive correlation between PIIINP and CRP (r=0.327; p 0.004) & WBCs (r=0.260; p0.024) and their ability studied by ROC curve for prediction of sepsis and showed AUC for PIIINP =0.883 with combined sensitivity and specificity of 88% and 82% with cutoff point of 103.3ng/ml, AUC for CRP=0.986 with combined sensitivity and specificity of 88% and 99.2% with cutoff point of 10.22 and AUC for WBCs=0.880 with combined sensitivity and specificity of 81% and 56% with cutoff point of 6.9.Zakynthinos et al [15] reported that  AUC for PIIINP was 0.96, C-reactive protein, and leukocyte count also exhibited high diagnostic accuracy to distinguish control subjects from patients with septic conditions with AUCs 0.99, and 0.91, respectively.

5- There was missing subgroup analysis. Did the accuracy of PIIINP predicting sepsis be similar in different age? in different infection types (UTI, pneumonia, FUO)? in different pathogen (GNB, GPC….)?

Answer (5): Yes, Sir. The accuracy of PIIINP predicting sepsis be similar in different       age, this can be explained by we did correlation between PIIINP and age and found no correlation p value (0.074) shown in table 4.  Regarding to different infection types, PIIINP is increased with resultant sepsis regardless of primary origin of infection (respiratory, or renal or cardiac, etc...).

6- Did the reference range of PIIINP not be age-adjusted????

  Answer (6): This study is a case -control study, we selected the control group matched age and sex with patient group to know reference range so we no need for doing age-adjustment.

7- In the section of discussion, it is necessary to report also all the new studies of the literature on this topic which are now missing. 

Answer (7): Yes, Sir, I added the new studies of the literature in my manuscript.

8- The mechanism of pathophysiological links in this field is reported in many previous studies. We suggested authors to report current pathogenesis in article.

Answer (8): Yes, Sir. We already reported current pathogenesis in manuscri.pt by red line as below;

The increased serum PIIINP levels in septic patients may relate to increased liver synthesis or decreased uptake by liver cells because procollagen propeptides are eliminated by the liver. In addition, although small amounts of PIIINP are excreted by the kidneys, acute renal failure is associated with increased synthesis of type III collagen [15].

9- Wu suggested author to provide a figure to overview results in this articles to make reader easily understand.

Answer (8): Yes, Sir.  If this figure is suitable, I can add it in the manuscript.

Round 2

Reviewer 1 Report

Dear authors,

Thank you for the revision. Most of my questions are solved. I am looking forward to your contribution to sepsis diagnosis in pediatrics. However, I have few minor concerns,

  1. I noticed that the capitalization in your title is not consistent. 
  2. Please make figure 2 clearer.
  3. Please briefly mention why did the authors only focus on PIIINP not "PINP"? Just tell readers the difference between them in the disease courses.

Author Response

Comments and Suggestions for Authors

Dear authors,

Thank you for the revision. Most of my questions are solved. I am looking forward to your contribution to sepsis diagnosis in pediatrics. However, I have few minor concerns,

Comment 1: I noticed that the capitalization in your title is not consistent. 

Answer (1): Yes, Sir. I corrected the capitalization in our title, as below:

Relation of Procollagen Type III Amino Terminal Propeptide Level to Sepsis Severity in Pediatrics

Comment 2: Please make figure 2 clearer.

Answer (2): Yes, Sir. I made figure 2 clearer as below:

Comment 3: Please briefly mention why did the authors only focus on PIIINP not "PINP"? Just tell readers the difference between them in the disease courses.

Answer (3): Yes, Sir. I added that in the manuscript as below:

N-terminal propeptide of type I procollagen (PINP) are recommended as markers of bone resorption and bone formation markers, respectively, correlated with corresponding histomorphometric parameters of bone formation and resorption [15]. So, we focused on PIIINP not "PINP.

15- S. Vasikaran, R. Eastell, O. Bruyère et al., “Markers of bone turnover for the prediction of fracture risk and monitoring of osteoporosis treatment: a need for international reference standards,” Osteoporosis International, vol. 22, no. 2, pp. 391–420, 2011.View at: Publisher Site | Google Scholar
